# A robust normalized local filter to estimate compositional heterogeneity directly from cryo-EM maps

Björn O. Forsberg ®[1,2] ✉, Pranav N. M. Shah ®[2] & Alister Burt[3]

Cryo electron microscopy (cryo-EM) is used by biological research to visualize biomolecular complexes in 3D, but the heterogeneity of cryo-EM reconstructions is not easily estimated. Current processing paradigms nevertheless exert great effort to reduce flexibility and heterogeneity to improve the quality of the reconstruction. Clustering algorithms are typically employed to identify populations of data with reduced variability, but lack assessment of remaining heterogeneity. Here we develop a fast and simple algorithm based on spatial filtering to estimate the heterogeneity of a reconstruction. In the absence of flexibility, this estimate approximates macromolecular component occupancy. We show that our implementation can derive reasonable input parameters, that composition heterogeneity can be estimated based on contrast loss, and that the reconstruction can be modified accordingly to emulate altered constituent occupancy. This stands to benefit conventionally employed maximum-likelihood classification methods, whereas we here limit considerations to cryo-EM map interpretation, quantification, and particle-image signal subtraction.

Protein, DNA, and other molecular polymers sustain the fundamental processes of life, and structural biology is the study of their functions and interactions. Cryo-electron microscopy (cryo-EM) aims to visualize them by aligning and averaging noisy images of many individual macromolecules[1-3], producing a 3D scalar field known as a map or reconstruction. The reconstruction represents the local density or scattering potential of the atoms that make up the imaged macromolecule, but other representations also exist[4-9]. The ability to confidently deduce the molecular structure from the reconstruction crucially depends on its local quality, which varies due to variability or heterogeneity among the images used to build the reconstruction[10]. Methods for isolating homogeneous subsets of particle images through e.g. clustering are therefore employed[11-15], and commonly utilize gradient descent methods[14,16,17]. Methods that parameterize the data have also been developed[7,18-21]. The ability to separate data computationally by any method affords cryo-EM an unprecedented capacity to analyze the experimental particle distribution[22,23], but this ability can be sensitive to user input. Existing alignment and clustering methods e.g. require a good initial (input) reconstruction to converge with good fidelity, since it provides sufficiently accurate estimates of image data parameters. However, lasting and potentially detrimental reference bias or over-fitting may occur[24]. This can distort the reconstruction and lead to misinterpretation or false features[25], and is exacerbated by the low signal-to-noise ratio (SNR) of cryo-EM data. So-called ab-initio 3D reconstructions can now be made without user input bias[14,26,27], but it is important to note that this nonetheless incurs downstream reference bias[24]. Current methods thus implicitly balance useful reference bias that permits convergence of the alignment against minimizing reinforcement of any reference bias arising from either spurious correlations in the data or the initial reconstruction. The imperative to minimize over-fitting has led to methods that avoid detrimental reference bias[28-30], and some which implicitly isolate useful reference bias[31,32] in a supervised fashion. Higher quality data and more robust reconstruction methods have also led to more frequent

[1]Department of Physiology and Pharmacology, Karolinska Institute, 171 77 Stockholm, Sweden. [2]Division of Structural Biology, University of Oxford, OX3 7BN Oxford, UK. [3]MRC Laboratory of Molecular Biology, Cambridge CB2 0QH, UK. ✉e-mail: bjorn.forsberg@ki.se

iterative updates to the reconstruction than previously, based on smaller subsets of the data. This has implicitly increased reference bias within the reconstruction procedure by giving descent methods higher inertia. It is thus appropriate and more prudent than ever to investigate if and how its outcomes could be improved through supplementing relevant reference bias based on the heterogeneity of the reconstruction. However, until now no method has been published to estimate the heterogeneity of the reconstruction, which states could be inferred to exist in the data based on the final reconstruction and its heterogeneity, nor which states can be extrapolated from the initial reference during clustering.

Recently established methods assign a measure of heterogeneity to the clusters[23,33,34], attain more efficient or complete clustering[35,36], and systematically choose an appropriate number of clusters[22]. Convergence of conventionally employed clustering is however largely subjective, and heterogeneity in the data may remain unresolved despite the apparent convergence of a given clustering algorithm[37]. As a result, biologically relevant differences may not be apparent across established clusters, and local reconstructions may suffer from undue incoherent averaging.

The present work formalizes the notion that cryo-EM reconstructions contain local information about latent heterogeneity[38]. Heterogeneity leads to local attenuation of the reconstructed density, a property we refer to as local scale. We provide OccuPy as a tool to estimate this local scale at all points within a reconstruction. OccuPy estimates local scale differences of arbitrary origin, e.g. due to flexibility, misalignment, and/or partial occupancy. OccuPy also provides a method to reduce the influence of the former, to better approximate macromolecular occupancy generalized as a scalar field (see "Discussion" section). This so-called occupancy mode is established by application of a low-pass filter to approximately neutralize the influence of blur variation on differential local contrast. This isolates composition heterogeneity of the reconstruction, which can justifiably be modified to emulate reconstructions expected from more homogeneous image data. Since convergence of e.g. data clustering is directed by reference bias for the cluster reference, it is also justified to consider that modifications of the latter based on estimated heterogeneity might be useful to improve convergence[34,39]. OccuPy requires only a reconstruction as input and runs in seconds without the need for GPUs or HPC infrastructure. The approach is thus possible to integrate into current cryo-EM processing pipelines based on both clustering and machine learning.

In this work, we establish the necessary formalism and tools to aid visual analysis of cryo-EM reconstructions, estimate local heterogeneity, and use it to improve current procedures. A GUI is provided for ease of use (Supplementary Fig. 1), expanding the toolkit for reconstruction analysis available to cryo-EM researchers.

## Results

### Local scale is accurately estimated against synthetic data

To evaluate if local scale can accurately estimate contrast degradation, we utilized simulated data with induced contrast degradation. A molecular model of malate dehydrogenase (PDB-1uxi, Fig. 1a) was altered by decreasing the occupancy of chain A, leaving all other atoms at full occupancy. Maps were generated based on this atomic model, using the theoretical electron scattering factors implemented in Gemmi[40], and the local scale was finally estimated from such maps. As evident in Fig. 1b, partial occupancy is qualitatively well estimated in the absence of flexibility or other sources of variable resolution. Systematic investigation also shows that when resolution is homogeneous, local scale quantifies local occupancy accurately (Fig. 1c). The application of a low-pass cutoff at 6 Å to establish an occupancy-mode estimate in this case reduces accuracy, ascribed to delocalization of the reconstructed detail which introduces voxel correlation that leads to a subsequent reduction of the effective sampling within the local

scale kernel window. Such a reduction in sampling under the established method will result in a reduced percentile $\tau$ (see methods for details). With a priori known occupancy, we can establish an empirical value of $\tau$ that results in more accurate occupancy estimation by a semi-exhaustive search at a fixed occupancy of 0.5 (Supplementary Fig. 2a), which thus constitutes a proxy of the voxel correlation within the local scale kernel window. This effective value $\tau_{eff}$ cannot be determined this way in general of course, but it is illuminating to do so in this analysis. It is observed that this accounts for the incurred pixel correlation, and improves the accuracy of chain A occupancy at all tested occupancies (Fig. 1c), despite being determined at a single occupancy of 0.5.

Small-molecule (ligand) occupancy is of broad interest to quantify and enrich in biological structures, so we also investigated if OccuPy can do so accurately with a small enough filter window to segment the ligand from its binding pocket. The analysis was thus repeated, instead modulating the occupancy of the NAD co-factor of PDB-1uxi (1d). It is evident that the surrounding does influence attainable granularity of the scale estimate, tending it towards over-estimation, in particular at lower ligand occupancy. Reducing the kernel size does mitigate this effect, and illustrates that the kernel size need only encompass a few voxels without significant detriment to the estimate when the fidelity and sampling of the underlying data is sufficiently high.

We also evaluate the capacity of the occupancy mode to neutralize local differences in resolution which might otherwise skew occupancy estimate. The occupancy of chain A of PDB-1uxi was thus fixed at 0.5 and the isotropic B-factor of all its atoms were modulated in the range 25–400 Å$^2$ (Fig. 1e). It is evident that the occupancy mode drastically reduces the B-factor dependence of the scale estimate, as intended. The remaining dependence is due to the lowpass filtration itself, which causes reduced scale due to delocalization of signal outside the kernel window. It is thus evident that OccuPy provides a reproducible and robust estimate, with some limitations. To illustrate this directly, the occupancy of all atoms of one NAD co-factor in PDB-1uxi was set to 0.4, leaving all other atoms at full occupancy. The density generated clearly shows that this co-factor is not visible at the same threshold as other elements (Fig. 1f). The same co-factor becomes evident following occupancy estimation and subsequent amplification, without being unduly exaggerated (Fig. 1g). Taken together, OccuPy is able to estimate their local scale in a meaningful way, but variations in resolution pose a challenge to accurate estimation. Occupancy mode does decrease the dependence on resolution but the theoretically derived values of $\tau_n$ neglect pixel correlation introduced by it, which thus tends OccuPy towards under-estimation of occupancy with increasing low-pass filtration.

### Occupancy estimated from noisy real data without an atomic model

In practice, real cryo-EM data is dominated by noise and ground truth of the underlying particle distribution itself is not known. To validate OccuPy in this setting, we first evaluate the RMS difference in local scale across half-set reconstructions of all EMDB-entries in Supplementary Note 1 of the Supplementary information, where this was available. As expected, the consistency is variable subject to the inherent noise of the reconstruction(s), which correlates so strongly with resolution that the latter determines the consistency of the estimated local scale almost entirely. Conveniently, the relative inherent uncertainty of the local scale estimate due to noise expressed in percent appears to be effectively proportional to the resolution in Å (Supplementary Fig. 3). For most published maps (which are better than 5 Å) this implies an uncertainty below 5% relative error in the local scale estimate.

Next, we utilized a set of particle images that have been aligned, symmetry-expanded, and signal-subtracted to visualize the rotavirus spike protein, which has partial occupancy on average. We estimate

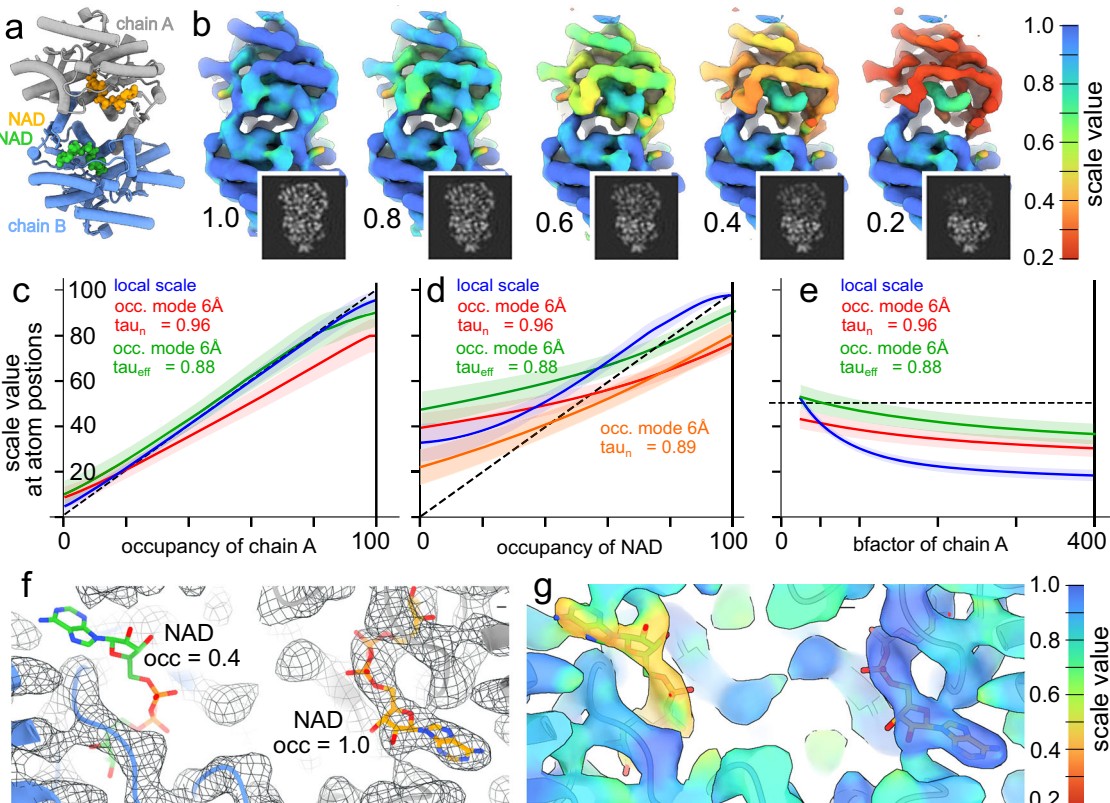

**Fig. 1 | Local scale estimation of simulated data. a** Malate dehydrogenase dimer PDB-1uxi, with chains colored individually. Two NAD co-factors are also shown in ball-and-stick representation. The occupancy of atoms in chain A was reduced and the local scale estimated from synthetic maps generated from this atomic model. Maps were also amplified with a power of 30. **b** Amplified maps, annotated by the occupancy of chain A, and colored according to the estimated local scale. A gray-scale volume slice of the input is also shown (insets). The kernel size and radius used were 5 and 2.5 pixels (4.3 Å and 2.15 Å), respectively. **c** Average local scale map value at atom positions of chain A, as a function of occupancy. The shaded area represents the standard deviation. The dashed line denotes ideal estimation. **d** as panel **e**, but for the NAD co-factors. Orange is equal to that of the red, using a reduced kernel size and radius of 3 and 1.6 pixels respectively, and auto-calculated $\tau_n = 0.89$. **e** as panel **c**, but as a function of the b-factor of atoms, at occupancy 0.5. **f** The density generated by reducing the occupancy of atoms in one NAD co-factor to 0.4. **g** The density in panel **c**, following amplification based on occupancy-mode local scale, also colored according to the estimated scale.

the spike foot occupancy of reconstructions using successively reduced numbers of random images from this set, to investigate robustness of the occupancy estimate to reduction in SNR and orientation coverage. The same procedure was conducted for a higher occupancy subset of the images, selected by conventional classification in RELION. To allow targeted evaluation of the spike foot in the absence of atomic assignment and at resolutions where atomic assignment is not possible, we designate an auxiliary custom scale estimation kernel (OccuPy option –target-mask), which derives a custom dedicated $\tau$ percentile independent of the global estimation parameters. We find that increased noise tends local occupancy towards slight over-estimation in OccuPy (Supplementary Fig. 4a). This is rationalized by the use of a max-value filter as the primary contrast metric within OccuPy, and highlights that while OccuPy employs rigorous adaptive methods to assign full occupancy, it is more sensitive to error when the point of null occupancy is ambiguous. OccuPy implicitly defines null occupancy based on the assumption that input images have been conventionally normalized against background. This assumption ensures that the estimated occupancy is not affected by the inherent noise nor the accuracy of the estimated noise-model (solvent model). However, it is evident that under elevated noise the occupancy mode local scale is over-estimated, which is more noticeable in regions where the occupancy is low and thus approaches the noise distribution (Supplementary Fig. 4a). To remedy this, OccuPy includes the option to re-calibrate the zero-point scale to the point where confidence exceeds that of the noise model, termed noise-level recalibration. However, at low SNR this is observed to instead risk an

under-estimate the occupancy-mode local scale where the solvent peak does not significantly depart from the regions of interest (Supplementary Fig. 4b). It is therefore advisable to combine this with the use of a solvent-definition that delineates and accurate solvent model in high-noise settings. The latter is also observed to largely compensate for the under-estimation incurred by noise-level recalibration of occupancy-mode local scale (Supplementary Fig. 4c). The implications of applying the noise-level recalibration (or not) is further discussed later.

This evaluation shows that the asymptotically derived local scale can be relatively accurately estimated in the prescience of significant noise. Asymptotic occupancy of 0.2 can e.g. be confidently estimated within ± 0.1, based on a reconstruction using as few as 1000 particles. The variance of the occupancy estimate in this analysis does exceed that expected from sampling of a Bernoulli random variable at the given asymptotic probability alone, which is attributed to the noise and variations in Fourier completeness.

## Modification by local scale emulates homogeneous data
The present work has devised methods to modify a reconstruction proportionally to the estimated local scale by amplification or attenuation of partial occupancy, as described in the methods section and exemplified in Fig. 2 and Supplementary Fig. 5. These modifications will at most remove or equalize partial occupancy, but can be adjusted to decrease the extent of modification. Consequently, these modifications at full or finite power are natural methods to modify the reference bias in numerous current cryo-EM processing tools

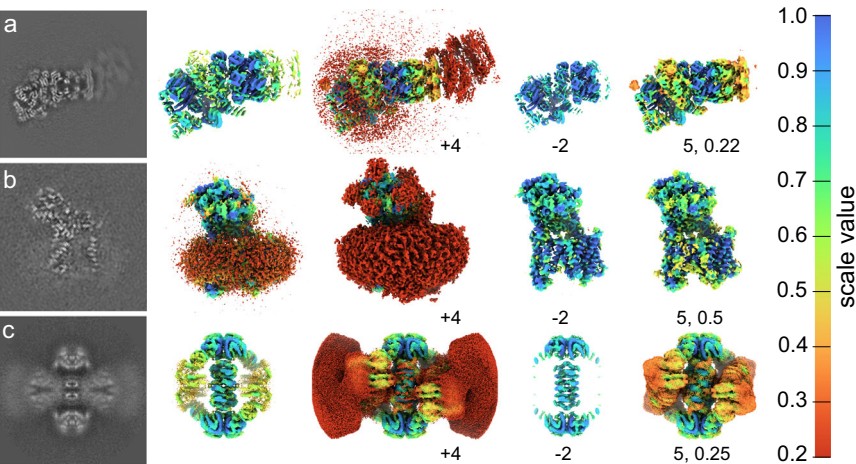

**Fig. 2 | Modification of reconstructed density by local scale.** All isosurface representations of each EMDB entry utilize the same threshold value for representation. **a** EMD-14085 shown as a central slice, and iso-surface representation colored by estimated local scale. Further, it was amplified ($\gamma = 4$), attenuated ($\gamma = 2$), completely or partially, respectively. The GUI also permits a sigmoid and sigmoid-modified ($\gamma = 5, \mu = 0.22$). **b** EMD-3061, as in panel **a**, with sigmoid pivot $\mu = 0.5$. **c** EMD-31466, as in panel a, with sigmoid pivot $\mu = 0.25$. Central slices of all modified reconstructions are also shown in Fig. 5.

completely or partially, respectively. The GUI also permits a sigmoid modification that combines amplification and attenuation (Supplementary Fig. 1), tailored to user-interactive visualization and modification such as subtraction. We therefore evaluate how amplification, attenuation, and differential sigmoid modification of local scale manifest in reconstructions with features encountered during cryo-EM processing, including heterogeneity, flexibility, misalignment, and amorphous regions such as inherently disordered detergent. First, EMD-14085 displays partial constituent occupancy and limited flexibility. As a result, we might expect the occupancy-mode local scale to be accurate but under-estimate macromolecular occupancy (Fig. 2a). In line with this, amplification restores low-occupancy components, but also amplifies some inherent noise. In part, this noise is elevated in an apparently spherical region which was presumably used as a mask for classification to establish EMD-14085. The use of fully amplified reconstructions for enforcing reference bias should thus be used with caution or under due noise reduction, through e.g. subsequent low-pass filtering. Conversely, attenuation acts conservatively at under-estimated scale, and does not suffer any detrimental noise amplification effects. Sigmoid modification with a tuned pivot value achieves amplification without undue noise amplification, but omits components with very low occupancy.

Second, we apply modifications to EMD-3061 (Fig. 2b), which has a region of detergent surrounding a transmembrane protein[41], since such regions are conventionally subject to signal subtraction[31] to reduce their influence on particle alignment and classification. Their resolution is ill-defined[42]; their physical extent can be determined with quantifiable accuracy, but any internal structure is effectively infinitely poor. Further, such a region is expected to have full occupancy since de-solvation of the transmembrane region is crucial for its structure, but its amorphous nature results in incoherent averaging that reduces local scale. This is also observed; the detergent micelle displays reduced local scale. Because local scale does not represent occupancy in this case, it can not be compensated by direct filtering to emulate decreased heterogeneity. However, regions that display reduced local scale due to incoherent averaging may still be suitably modified for visualization and induced reference bias. Amplification in this case leads to grave exaggeration of local mass since local scale is severely underestimated due to resolution effects, even in occupancy-mode. This also displaces the reconstruction gray-scale outside the expected range, which further reduces its fidelity as the expected reconstruction from more homogeneous data. This again advocates that amplified

reconstruction may be unsuitable for direct interpretation or use without further considerations. In the case of EMD-3061, attenuation and sigmoid modification curiously also cause undue modification, since no part of the input data is expected to be without a detergent micelle. The utility of such modification is however evident from its use in existing protocols for signal subtraction to reduce the influence of regions that cannot be coherently aligned, permitting structured regions to be better resolved. In this capacity, attenuation and sigmoid modification both appear well suited, signifying a direct way to weight reconstruction data by an objectively determined local property. To corroborate that such an approach is more broadly applicable to e.g. macromolecular flexibility, we subject EMD-31466 to the same analysis (Fig. 2c). This displays the same tendencies as EMD-3061, showing that local scale can be used to (de-)emphasize local regions of reconstructions in an automated or semi-automated manner using an objectively estimated attribute.

## Variations in local mass can be accommodated

OccuPy is not equipped to consider the expected average density, charge, mass, or other causes for altered scattering potential of constituent atoms, nor a physical image formation model. Instead, it assumes that all regions with equal resolution and full occupancy will produce an identical voxel intensity distribution. This does not hold for atoms of unequal mass, which could potentially be estimated at a different scale. To evaluate how robust OccuPy is to such situations, we first considered the case of a nucleosome protein-DNA complex (EMD-32148), since the phosphate-rich backbone could lead to underestimated scale of protein components. This does not seem to be the case (Fig. 3a). Next, we examined a high-resolution (1.22 Å) reconstruction of apoferritin (EMD-11638). First, the region **W** used normalize the scale estimate was reduced to a single pixel, which makes the local scale estimate sensitive to single pixels with high values, e.g. at high-mass atoms. Methionine side-chain sulfur atoms are thus estimated at higher scale than surrounding protein (Fig. 3b). The default size of the region **W** (Eq. (2), methods) efficiently diminishes the influence of such heavy atoms by considering the max-value distribution of the highest contrast region found (Fig. 3c–d), unless it dominates an entire region **W**. This nevertheless emphasizes that full scale (or occupancy) is a relative term in the absence of true underlying mass, here defined based on the size of **W** and the percentile $\tau$. A smaller region **W** allows local mass to define full scale. Conversely, a larger **W** will reduce the influence of local mass differences. However,

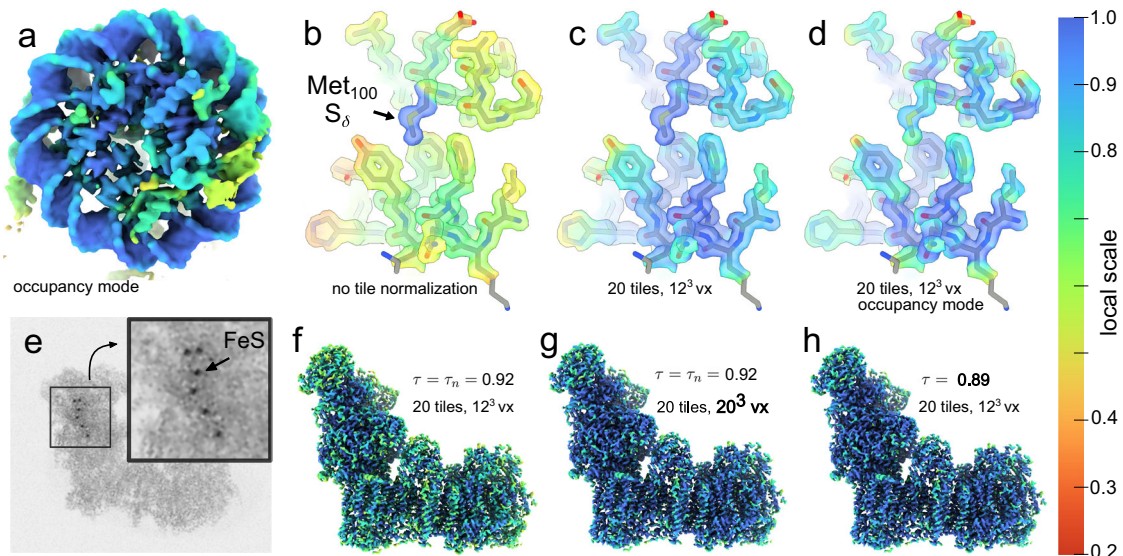

**Fig. 3 | Local scale estimation under variation of atomic mass. a** EMD-32148 shows no difference in estimated scale of DNA relative to protein. **b** local scale of EMD-11638 without normalizing region **W** shows sensitivity to local mass. **c** local scale of EMD-11638 with default parameters. **d** occupancy-mode local scale of EMD-11638 with default parameters. **e** Maximum-value projection of the EMD-13611, showing FeS-clusters. **f** local scale of EMD-13611 with default parameters. **g** local scale of EMD-13611 with larger normalizing region **W**. **h** local scale of EMD-13611 with reduced percentile $\tau$.

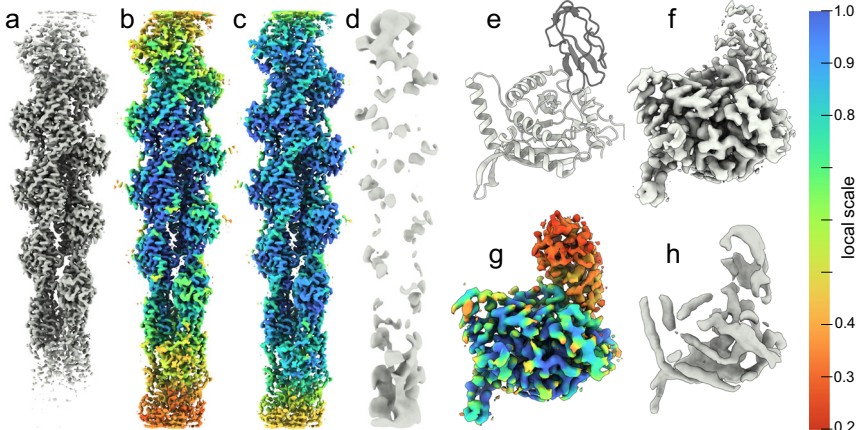

**Fig. 4 | Local scale estimates in the presence of flexibility or misalignment. a** Input map (EMD-30171). **b, c** Amplified map, colored according to estimated local scale and occupancy-mode local scale, respectively. Panels **a**–**c** are shown at the same threshold value. **d** Low-pass filtration of the amplified map, showing exaggerated mass in low-scale regions. **e** The PDB-7b9k atomic model, with the catalytic trans-acetylase domain (CTD) in white and a lipoyl domain (LD) shown in black. **f** The cryo-EM reconstruction EMD-12104, showing apparent partial occupancy with negligible flexibility. **g** The amplified reconstruction, utilizing and colored by the occupancy-mode local scale. **h** Low-pass filtration of the amplified map.

setting **W** too large may cause systematic overestimation of the scale since no region of this size will be uniform at full scale. The default size of **W** in OccuPy is chosen to consider the size granularity of biomolecular complexes typically reconstructed by cryo-EM, while neglecting individual atoms. Validation of the tile size was considered in the synthetic data validation (Fig. 2b, and can be adjusted by the user. To further examine the potential pitfalls of the method, we also estimate the scale of respiratory complex I (EMD-13611), which contains a number of FeS clusters (Fig. 3e). Naturally, these FeS clusters are estimated at full scale. In spite of this, the protein content is only slightly under-estimated (Fig. 3f). A decreased value of $\tau$ can further compensate for this (Fig. 3g), and the size of the region W may be increased to define full scale (Fig. 3h). Both these parameter adjustments reduce the influence of high local values, but the latter offers a direct interpretation as redefining the granularity of the estimate through the size of the reference region **W**.

## Macromolecular flexibility can be partially accommodated

Local resolution variability within published reconstructions is common and primarily due to flexibility[43] and other sources of misalignment[24]. We therefore evaluate OccuPy against a a reconstruction of a flexible helical assembly (F-actin EMD-30171) for which negligible variation in occupancy is expected (Fig. 4a). In line with expectation, the local scale correlates with decreased resolution further from the box center (Fig. 4b, c). The local scale in occupancy mode is less affected, but still indicates decreased occupancy, which is not ideal. The estimated occupancy can be validated by performing full amplification (power $\gamma = 30$), followed by a low-pass filtration. Doing so for EMD-30171 reveals that mass has been exaggerated in such regions (Fig. 4d), indicating that the occupancy was strictly under-estimated. This can not be ameliorated by increasing the low-pass cutoff, in line with expectation (see "Methods" section). To contrast these findings, EMD-12104, exhibits

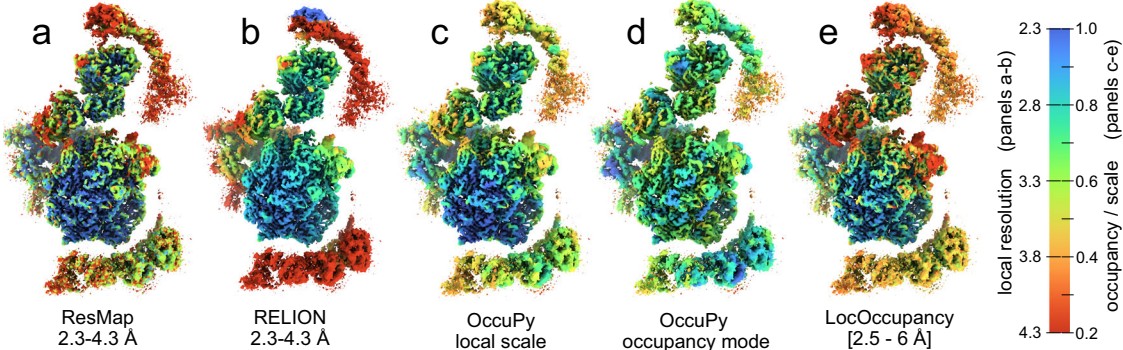

**Fig. 5 | Comparison of local resolution and occupancy estimation methods.** **a** The local resolution of EMD-13015 was estimated using ResMap[44] in the range 2.3–4.3, with a resolution step of 0.25 Å. **b** The local resolution was estimated using RELION[13] with a spacing of 10 Å. **c, d** The local scale and and occupancy-mode local scale estimated using OccuPy. **e** The occupancy estimated using LocOccupancy[45], using a resolution range of 2.5–6 Å.

partial occupancy but negligible flexibility, in which case low-pass filtration of the amplified map shows no emphasis on the regions estimated at low occupancy scale (Fig. 4h), indicating accurate occupancy estimation and appropriate amplification. To further illustrate the fidelity of the local scale estimate as compared to existing methods, EMD-13015 was used. The local resolution varies as estimated by ResMap[44] and RELION[13] (Fig. 5a, b). The local scale reproduces the relative contrast estimation (Fig. 5c). In occupancy-mode, the local scale is more homogeneous, indicating that these differences can be attributed largely to flexibility (Fig. 5d). The occupancy estimated by LocOccupancy[45] (Fig. 5e) however correlates strongly to the estimated resolution, when provided with the range of spatial frequencies over which resolution is deemed to vary by consensus methods. Other ranges were able to reduce this effect, notably the omission of the range where resolution varies (Supplementary Fig. 6). This indicates that LocOccupancy suffers the same vulnerability to variation in resolution as OccuPy, and that the implicit solution is similar to that advised in OccuPy.

## Local scale can be estimated from sharpened maps

Maps are typically post-processed after reconstruction to maximize information and fidelity. Global B-factor estimation and compensation is most common[46], but local filtering[47–49] and machine-learning[50] are also employed. To investigate if post-processing introduces or obscures expected features to the detriment of a faithful scale estimate, we modify EMD-3943 by common post-processing methods and then estimate local scale, since this reconstruction contains differential resolution in its subunits and partial occupancy of a bound recycling factor (RRF). First, the local scale estimate of a map sharpened by a global B-factor exhibits a larger range (Supplementary Fig. 7d), since local contrast is increased in proportion to SNR. The local scale estimate in occupancy mode is however similar to that estimated from the unmodified map. While OccuPy is not intended to be used on post-processed maps, it nonetheless appears permissible. Next, a local-resolution filtered reconstruction displays decreased local scale in the subunit at lower estimated resolution. This reconstruction is however highly similar to the original map, both in terms of the local scale estimate and that in occupancy mode (Supplementary Fig. 7g–i). Finally, a reconstruction modified by deepEMhancer[50] shows a very uniform full scale, apart from the RRF, which is likely at a lower occupancy. Indeed the RRF is lower in occupancy-mode as well. Curiously, the subunit occupancy scale is inverted with respect to the unmodified input map (Supplementary Fig. 7k), indicating that DeepEMhancer alters local mass dependent on the local resolution. Based on this, we surmise that reconstruction post-processed by machine-learning methods are not suited for use in OccuPy without further prior validation.

## Improved robustness of confidence estimate through solvent definition

OccuPy estimates a solvent model and subsequent confidence map to avoid solvent noise amplification. Such a confidence map assigns a value to each voxel, signifying the probability that it something other than solvent, which can be considered a soft solvent mask. In some cases this solvent model is incorrectly estimated due to unexpected solvent characteristics, in which case an additional input mask can be provided to limit the regions of the input map considered when determining the solvent model. We denote this a solvent definition, since it does not mask the output. We illustrate its use on the map of the asymmetric unit of a viral capsid re-framed such that the solvent volume is only 22% of the cubic map volume (42% of the map radius sphere) (Supplementary Fig. 8a). The capsid interior also contains a disordered component with low variance but higher mean than the solvent. An accurate solvent definition (Supplementary Fig. 8b) that excludes all protein content and capsid interior results in a single Gaussian solvent peak and accurate confidence. If the viral capsid spike and interior is not excluded by the solvent definition (Supplementary Fig. 8c), the solvent model and confidence is still accurately estimated. (Supplementary Fig. 8f). This demonstrates that the solvent definition does not strictly enforce what is amplified, permitting map modification outside the provided solvent definition.

## Discussion

This work defines local scale as an estimate of relative contrast in cryo-EM reconstructions, which is assumed to be proportional to heterogeneity in the data used. In the absence of flexibility we further interpret this as occupancy, signifying a mixing parameter of binary composition inherent to the input data. This is consistent with the accepted definition of occupancy in structural biology, where it annotates the relative occurrence of atoms in a model that best agrees with the map on which it is based, whereas our interpretation annotates the map itself. This permits quantification where atoms cannot be distinguished, but also leads to ambiguities in regions of partial disorder. It should thus be clarified that field-annotation of occupancy as a mixing parameter of binary origin is only relevant to the extent that the underlying heterogeneity is in fact binary. The local scale is a natural generalization of occupancy (to heterogeneity more broadly) where the origin is non-binary. It is nonetheless informative to attempt to decompose the local scale as originating in either binary or continuous heterogeneity. The occupancy-mode local scale implemented here attempts to omit the latter to render the local scale maximally interpretable as a mixing component of binary heterogeneity, but depending on the nature of the underlying heterogeneity this may not be possible without ambiguity. This should be considered in interpreting the local scale output by OccuPy.

We go on to demonstrate local scale can quantify macromolecular occupancy in cryo-EM reconstructions without a molecular model, and be used as a meaningful means to modify them. However, accurate estimation of local scale may require parameter tuning response to reconstruction characteristics, which could lead to user confirmation bias. LocOccupancy[45] is the only other method designed to approach quantification of compositional variation, meriting a direct comparison. LocOccupancy requires a resolution range to be specified which in some sense dictates the granularity of the estimation, whereas OccuPy instead estimates local properties with a minimal kernel given the resolution of the reconstruction, and regulates the granularity of the occupancy estimate through the normalization region **W**. Both LocOccupancy and OccuPy are also dependent on a percentile cutoff, which signifies different characteristics in each implementation. LocOccupancy sets this value to 0.25, signifying the top percentile that defines full occupancy in some sense, whereas OccuPy automatically sets a theoretically optimal value to minimize the probability of both over- and under-estimating the occupancy. In further comparison, LocOccupancy naturally maps each region to the [0, 1]-range, whereas OccuPy instead normalizes by a value lower than the global maximum and clamps the estimate to the [0, 1]-range. A clear benefit of LocOccupancy is that it effectively marginalizes the occupancy estimate over the desired resolution range, while OccuPy makes no such provision and estimates occupancy-mode local scale under the assumption that variations in local resolution have been neutralized. Despite this apparent disadvantage, Fig. 5 suggests that OccuPys occupancy-mode is able to disregard resolution-dependent contrast degradation better. Fig. 5 also shows that the local scale is an accurate estimate of relative local quality, which is tantamount to resolution. Resolution is however a contested term in cryo-EM[42,51]. By consensus, the spatial frequency at which the global Fourier shell correlation (FSC) drops below a given significance[46] is quoted as the best resolution at which the reconstruction can be reliably interpreted. Local metrics also permit variations to be quantified under the term "resolution". As discussed elsewhere[52] these measures are not identical, and the term resolution is thus not well defined. It is however clear that resolution correlates positively with data amount and quality, and how coherently it can be averaged. This in turn principally depends on macromolecular flexibility and occupancy (as well as particle misalignment), which mirrors that of the local scale estimated here. In this sense, the OccuPy local scale does constitute a true estimate of relative local resolution. However, OccuPy assumes that the density originates from identical point scatters. Due to their variation in mass and occupancy, the local scale is not a universal estimator of resolution. When resolution becomes poorer than the physical spacing of the point source of scattering, their environment also influences the scale estimate, as shown in Supplementary Fig. 9. Bearing these points in mind we conclude that the local scale is an accurate estimate of the relative local resolution, but that this estimate is dependent on properties that e.g. FSC-based resolution estimates are independent of.

State-of-the-art cryo-EM processing attempts to parameterize or embed data in a neural net using machine learning approaches, which generalizes discrete classification. OccuPy finds further use in this context, where it could validate remaining latent heterogeneity in the resulting reconstructions, and provide intuitive quantification of the latent space. OccuPy can also supply labels when reconstructions in existing databases are used for training, or indeed direct scoring functions employed to train occupancy-aware machine-learning approaches. Amplification using OccuPy can also serve to equalize reconstructions to improve initialization of methods dependent on e.g. pseudo-atom fitting, since it reflects a more homogeneous map where all regions of relevant consideration appear more self-similar. OccuPy is thus not limited to visualization or discrete classification, but supplies a measure of heterogeneity that reflects natural variations in cryo-EM data that is

merited and possible to use for quantification and targeted consideration in any cryo-EM processing paradigm.

Taken together, we find that OccuPy is the only tool able to quantitatively estimate macromolecular occupancy within cryo-EM reconstructions and modify it in a meaningful way, but that user intervention may be necessary to assure fidelity in this process. Through its GUI (Supplementary Fig. 1), users can directly adjust estimation parameters such as input low-pass frequency, kernel size, and normalization tile-size, and visualize the results. The solvent model can also be directly evaluated, and the optional input solvent definition constructed. To permit easy integration with current processing pipelines, a command-line interface and python module is also provided. From this interface, further evaluation is also facilitated by invoking UCSF ChimeraX[53] with a command-script that is part of the default output. This will also display complementary visualizations to evaluate the results. OccuPy may also be used to improve signal subtraction by providing accurate subtraction masks, and its capacity to accurately estimate local scale with minimal user input suggests that this capacity could be employed in iterative refinement procedures, for which it is also the only viable method considering speed of execution (Supplementary Table 1). OccuPy thus stands to improve current procedures, where it could be used e.g. with a weak power to bias reference-based alignment and clustering, however, this remains to be validated in practice. OccuPy thus offers an example of how local spatial analysis can improve interpretation of cryo-EM reconstructions, which stands to be developed further to benefit future reconstruction analysis and refinement algorithms broadly.

## Methods

### A spatial filter to estimate local scale

The best-resolved region in a cryo-EM reconstruction displays the highest contrast. We axiomatically define the images used to make the reconstruction to be completely homogeneous with respect to this region. Globally homogeneous input data would thus result in a theoretically ideal cryo-EM reconstruction $F_{ideal}$. In $F_{ideal}$ all non-solvent regions exhibit identical local scale. In practice, contrast is attenuated through local flexibility and/or partial occupancy, as well as the inclusion of misaligned or bad particles. The observed map $F$ can thus be considered $F_{ideal}$ degraded by the local scale $S$:

$$F = F_{ideal} \cdot S \tag{1}$$

where $\langle \cdot \rangle$ denotes pixel-wise multiplication and $\{S | S_i \in [0, 1] \; \forall \; i\}$. $S$ is thus a normalized estimate of the local signal strength. We estimate $S$ using a windowed max-value filter over the local neighborhood $\mathbf{V}_i$ of each pixel $i$, i.e. by spatial filtering. This measures the width of the voxel value distribution sampled within $\mathbf{V}_i$, which signifies signal above noise in cryo-EM reconstructions. A max-value filter is fast, insensitive to the inclusion of solvent, and robust to noise even for very small window sizes. It is also capable of preserving sharp transitions, with respect to a morphological dilation.

To normalize the estimated scale at each pixel $i$, we have to determine the maximal expected value given the finite sampling in regions around $i$. To do so, we first subdivide the reconstruction into non-exhaustive regions $\mathbf{W}_j$. A percentile filter with parameter $\tau$ is used for robustness to high-value outliers, with due consideration to follow. The region $j$ with the largest such value is used to determine full scale at a given percentile $\tau$, and is used as a denominator to normalize the scale estimate:

$$\hat{S}_i = \frac{\max\limits_{i \in \mathbf{V}_i}(F_i)}{\max\limits_{\{\mathbf{W}_j\}}\left(\mathrm{per}_\tau\limits_{i \in \mathbf{W}_j}(F_i)\right)} \Bigg\rceil_{[0,1]} \tag{2}$$

where we have defined a function $\lceil$ which clamps values to a specified interval

$$x\lceil_{[a,b]} \equiv \max(a, \min(x,b)) \qquad (3)$$

since the max-value reduction over $\mathbf{V}_i$ may exceed that of the percentile $\tau$ in the full-scale region found. The established procedure permits the intensity distribution of a small region to define full scale/contrast, and does not enforce any specific portion of the reconstruction to be assigned any nominal scale value. It also obviates the need for masking areas of interest. The set of regions $\{\mathbf{W}_j\}$ need not be exhaustive since the denominator is globally defined. Occupy thus utilizes a sparse set of $j$ regions $\{\mathbf{W}_j\}$, evenly distributed across the reconstruction. By default, 8000 ($20^3$) regions of 1728 ($12^3$) voxels are used, which represents a fundamental granularity of biomolecular components that works in a broad range of test-cases. This is a tunable parameter in the present implementation.

We now derive a reasonable choice for the percentile $\tau$. We first note that as long as sufficiently many regions $\mathbf{W}$ are sampled, $\hat{S}_i$ may be an overestimate of $S_i$ by at most $1 - \tau$. On the other hand, $\hat{S}_i$ would instead be an underestimate if the number of elements $n_{\mathbf{V}}$ within $\mathbf{V}$ is unlikely to have sampled as high as the percentile given by $\tau$. As a compromise, we seek the percentile $\tau_n$ of the distribution that equals the confidence that the maximum value of n samples is also $\tau_n$. Considering the reconstruction values in a local region to be a random variable $X$, we solve

$$G_{Y_n}(x) = 1 - G_X(x) \qquad (4)$$

where $G$ is the cumulative distribution function (CDF) and $Y$ is the maximum value distribution

$$Y_n = \max\{X_1, \ldots, X_n\} \qquad (5)$$

However, the CDF of $Y_n$ can be simplified as

$$G_{Y_n}(x) = P(Y_n \le x) = \prod_{i=1}^{n} P(X_i \le x) = G_X(x)^n \qquad (6)$$

Consequently, we can rewrite Eq. (4) as

$$G_X(x)^n = 1 - G_X(x) \qquad (7)$$

from which we see that $\tau_n$ is simply the only positive real root to the polynomial

$$x^n + x - 1 = 0 \qquad (8)$$

This result is independent of the underlying distribution of $X$. However, the CDF of Eq. (6) is only separable under the assumption that adjacent voxels are independent. By setting $\tau$ in Eq. (2) to $\tau_n$ as dependent on the kernel size $n_{\mathbf{V}}$, we are thus guaranteed that $S_i$ at most over-estimated by $1 - \tau_n$, and under-estimated with a probability $1 - \tau_n$. By solving Eq. (8) we find that $\tau_n \ge 0.9$ for $n_{\mathbf{V}} \ge 27$. This places narrow error bounds on the scale estimate for any realistic kernel size. In reality, the voxels sampled within V are not independent, which reduces the effective sampling number compared to $n_{\mathbf{V}}$, such that $\tau$ should reasonably be set lower than $\tau_n$. Additionally, a region $\mathbf{W}$ is assumed to exist that has homogeneous and full scale. When this is not the case, the normalizing value in the denominator of Eq. (2) may be increased due to e.g. high-mass atoms in the full-scale region, or conversely decrease due to inclusion of solvent. This will lead to systematic under- and over-estimation of local scale, respectively.

Finally, we note that in OccuPy, the kernel size $k$ (voxels along each dimension) is automatically calculated in resolution-dependent

manner as the smallest odd integer larger than

$$k = 2 \cdot r/d \qquad (9)$$

where $r$ is the applied input low-pass filter or resolution of the input reconstruction and $d$ the input voxel size. The kernel size determines a radial cartesian kernel as illustrated in Supplementary Fig. 10.

### Establishing an occupancy-mode

We first note that $\hat{F}_{\text{ideal}}$ can be found as an estimate of $F_{\text{ideal}}$ as

$$\hat{F}_{\text{ideal}} = F \cdot \hat{S}^{-1} \qquad (10)$$

The instability of inverse filtering at low values of $S$ is handled by a complementary confidence estimation to follow. From Eq. (10), it is clear that $S$ constitutes a spatial filter capable of modifying the estimated macromolecular occupancy without masking or segmentation. The local scale $S$ is however a measure of contrast attenuation, which correlates with both resolution (peak broadening) and occupancy (peak reduction). Modification of a reconstruction through the use of $S$ as a spatial filter is however only appropriate to compensate or further exaggerate peak reduction due to occupancy, not resolution. To omit resolution-dependent effects, we employ the simple procedure of estimating the local scale from a low-pass filtered copy of the input reconstruction. We term this occupancy-mode local scale. The low-pass procedure achieves resolution-dependent attenuation by the same magnitude at all points in the reconstruction. All regions estimated must thus be affected by the low-pass filter, lest regions better resolved should be estimated at higher scale by virtue of higher resolution. In addition, we implicitly assume that local density values do not suffer differential influence by peak broadening of nearby values. This is violated e.g. comparing points internal to the protein core to those near solvent. This violation becomes more severe at lower resolution, so that any omission of resolution-dependent effects is countered by detrimental convolution of local contrast. The occupancy-mode local scale will thus be difficult to establish faithfully by the low-pass filter approach when the reconstruction displays large variations in local resolution due to flexibility and imperfect particle image parameters. The general local scale is however accurate with arbitrary internal variation in local contrast.

### Noise-level recalibration

OccuPy by default assumes that the image extraction performed background normalization such that solvent background has been globally defined to have zero mean. In this context, the zero-point local scale is defined at a reconstructed voxel value of 0, which is correct in the limit of no noise, and makes the local scale estimate independent of the noise model estimate. However, one may optionally recalibrate the zero-point occupancy to the upper noise level, such that voxel values which are equally likely to originate from the estimated noise model or not defines the zero-point of local-scale. To do so, the primary confidence limit $S_p$ is found, corresponding to the local scale where confidence drops below 0.5. The scale is then recalibrated as such:

$$S' = \frac{S - S_p}{1 - S_p} \qquad (11)$$

### γ-modification of estimated scale

To achieve attenuation or amplification or partial occupancy, Occupy implements a proportional and inverse power scaling of the estimated occupancy-mode scale $S$ by a power $\gamma$ analogous to conventional

$\gamma$-correction.

$$F_{+\gamma} = F \cdot S_{+}\gamma = F \cdot S^{1/\gamma - 1}\left( = \hat{F}_{ideal} \cdot S^{1/\gamma}\right) \qquad (12)$$

$$F_{-\gamma} = F \cdot S_{-}\gamma = F \cdot S^{\gamma - 1}\left( = \hat{F}_{ideal} \cdot S^{\gamma}\right) \qquad (13)$$

Amplification in this context is thus attenuation of $\hat{F}_{ideal}$ by $\hat{S}^{1/\gamma}$, signifying less heterogeneity than what was estimated from the input data. Attenuation is conversely application of more heterogeneity than $\hat{S}$. For consistency, the present implementation only permits $\gamma \geq 1$. $\hat{F}_{ideal}$ corresponds to $F_{+\infty}$, a direct inverse filter of the input by $\hat{S}$. $\gamma$-modification is illustrated in Supplementary Fig. 11a and exemplified in Fig. 2.

Attenuation of a reconstruction can be directly applied to attain precise and automatic masks for particle subtraction workflows[31]. Conventionally, a mask $M$ is provided where each voxel value $m \in [0, 1]$ determines the retention of image values:

$$I_{subtracted} = I - \mathbf{P}_{\phi}(F \cdot (1 - M)) \qquad (14)$$

where $\mathbf{P}$ is the projection operator and $\phi$ is the alignment of image $I$. Conventionally the mask is constructed by manual volume segmentation and user manipulation. Here, the estimated scale $S$ and the desired output scale $S'$ can be used to formulate an optimal mask $M$:

$$\hat{F}_{ideal} \cdot S' = \hat{F}_{ideal} \cdot S - (1 - M) \cdot \hat{F}_{ideal} \cdot S \Longleftrightarrow \qquad (15)$$

$$M = S'/S \qquad (16)$$

$M$ is on the interval $[0, 1]$ as long as $S' < S$, i.e. when the desired output scale $S'$ is lower than the input scale $S$. This signifies strict attenuation, which is reasonable as components are to be subtracted. Occupy provides a simple interface to create such a mask, with any necessary adjustments.

## Sigmoid modification

OccuPy also implements a sigmoid modification, which attenuates components below specified local scale value, but also amplifies components above the same value. Like $\gamma$-modification, sigmoid modification is dependent on the power $\gamma$. Again, $\gamma = 1$ signifies no change, and increasing values result in increased modification. The additional parameter $\mu$ signifies the threshold scale value that remains unmodified and is thus denoted the pivot value. Formally, the scale $S$ is altered as

$$S_{\gamma,\mu} = \left(1 + \left(\frac{S \cdot (1 - \mu_s)}{\mu_s \cdot (1 - \hat{S})}\right)^{-\gamma}\right)^{-1} \qquad (17)$$

where

$$\mu_s = \left(\left(\frac{1 - \mu}{\mu}\right)^{1 - \frac{1}{\gamma}} + 1\right)^{-1} \qquad (18)$$

The sigmoid modification is thus formulated as

$$F_{\gamma,\mu} = F \cdot S_{\gamma,\mu} \cdot S^{-1}\left( = \hat{F}_{ideal} \cdot S_{\gamma,\mu}\right) \qquad (19)$$

Like attenuation, sigmoid modification can be used to construct a mask for particle subtraction according to Eq. (16), under the provision that the sigmoid mapping is adjusted so that values above $\mu$ remain unmodified. Sigmoid modification is illustrated in Supplementary Fig. 11b, c and exemplified in Fig. 2f.

## Suppression of solvent modification

Amplification is a form of inverse filtering, which is sensitive to noise. OccuPy therefore estimates a solvent confidence map to avoid amplification of solvent. While local signal-to-noise ratio (SNR) is a reasonable estimate of the confidence in the reconstructed voxel value, it is too strict for the purposes here. More leniently, we establish the confidence as the relative probability of observing a given voxel value in content over solvent. To do so, we determine a solvent model Θ as a Gaussian fit to the main peak of the reconstruction histogram, much like previous methods[54], resulting in a confidence map for each voxel. This is exemplified in Fig. 2c. OccuPy does not identify solvent regions prior to this fit, but instead relies on the assumption that the majority of the reconstruction volume is composed of solvent and has a pronounced peak in the image histogram. If not, the solvent variance is typically over-estimated, leading to decreased confidence of low-scale regions. To permit more accurate fitting in these cases, a solvent definition can be supplied in the form of a mask that covers the non-solvent regions of the input reconstruction. This is not employed as a mask, but instead delineates the regions omitted when fitting the Gaussian solvent model. Consequently, it does not restrict the confidence map and permits scale modification outside the provided solvent definition. This is shown in Supplementary Fig. 8.

Formally, the confidence $C$ computed as the ratio of the probability that each voxel pertains to the solvent model or not:

$$C_i = \frac{P(A_i \in [F_i - \delta/2, F_i + \delta/2]|F)}{P(F_i|\Theta)} \qquad (20)$$

where $\delta$ is the bin of the reconstruction histogram. The spatial filter $C_i$ essentially constitutes a soft solvent mask that is applied to suppress solvent, without segmenting the reconstruction or enforcing a hard threshold.

$$Q_{\pm\gamma} = F_{\pm\gamma} \cdot C \qquad (21)$$

$$Q_{\mu,\nu} = F_{\mu,\nu} \cdot C \qquad (22)$$

## Retention of solvent

OccuPy intends to modify reconstructions to mimic the expectation if the input data were more homogeneous. Solvent should thus not be excluded as described by Eq. (21). OccuPy therefore utilizes the inverse confidence to retain the original solvent background.

$$R_{\pm\gamma} = Q_{\pm\gamma} + F \cdot (1 - C) \qquad (23)$$

$$R_{\gamma,\mu} = Q_{\gamma,\mu} + F \cdot (1 - C) \qquad (24)$$

Further to this, attenuated noise is compensated in proportion to the attenuation:

$$T_{-\gamma} = R_{-\gamma} + N \cdot (S - S^{\gamma}) \cdot C \qquad (25)$$

where $N$ is the noise generated to have the same distribution and spectral properties as the solvent of the input reconstruction.

## Reporting summary

Further information on research design is available in the Nature Portfolio Reporting Summary linked to this article.

## Data availability

All EMDB entries used for development of OccuPy are listed in Supplementary Note 1 of the supplementary information. PDB-1uxi was

used to generate synthetic test data. Source data for Fig. 1c–e and supplementary Figs. 2–4 are provided in a Source Data file. Synthetic densities (see "Results" section, Fig. 1 and Supplementary Figs. 1 and 2) and subset reconstructions (see "Results" section and Supplementary Fig. 4) are deposited in Zenodo 10.5281/zenodo.8229242. Source data are provided with this paper.

## Code availability

The software is publicly available at github.com/bforsbe/OccuPy, and pypi.org/project/OccuPy/. Instructions, tutorials, and measures for transparent reproducibility are hosted on occupy.readthedocs.io.

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

## Acknowledgements

The authors are grateful to those who provided feedback and functionality testing, including Dari Kimanius, Marta Carroni, and Loic Carrique. The authors are grateful to Jesse Hopkins at BioCAT for help with software packaging. The present study was funded by the Swedish Research Council grant 2020-06413 (B.O.F). The computational aspects of this research were also supported by the Wellcome Trust Core Award Grant Number 203141/Z/16/Z and the NIHR Oxford BRC.

## Author contributions

B.O.F conceived the project, led experimental design, devised methods and theory, developed the implementation, performed data analysis, wrote the paper, and provided funding. P.N.M.S. conceived the project, developed the implementation, and provided revisions to the method and manuscript. A.B. developed the implementation and provided revisions to the method and manuscript.

## Funding

## Competing interests

The authors declare no competing interests.
