## [Peer Review File · Nature Communications]

A robust normalized local filter to estimate compositional heterogeneity directly from cryo-EM mapsReviewers' Comments:

Reviewer #1:

Remarks to the Author:

Cryo-EM reconstructions often display variable local resolution: molecular density is better resolved in some parts of the reconstruction and worse in others. This can be caused by heterogeneity of the sample (both in composition, aka occupancy, and conformation, aka flexibility) and inaccuracies in the reconstruction process (misalignment of particle images or incorrect estimates of other parameters). Identifying these effects is helpful for characterising a reconstruction, and adjusting for them can potentially help to improve the quality of the reconstruction.

In this manuscript, Forsberg, Shah and Burt aim to both estimate and adjust for the effect of heterogeneity in a cryo-EM reconstruction through the development of a new algorithm and software tool that they call OccuPy. The authors have done well to develop a new algorithm that appears to be effective and is much faster than any existing alternative approaches. This is an important problem and they have made some useful progress. As they say, this is a first step and the real potential of their approach would be if it could be used automatically as part of an iterative refinement and reconstruction process. However I expect this initial development will already be quite helpful to the community, particularly for scientists with challenging data sets and complex workflows that rely on detailed classification and signal subtraction approaches.

The authors generally do a good job of describing their work and the results broadly support the conclusions. However I do have concerns about some aspects of how the work is presented and so I recommend revision before the manuscript is accepted for publication.

My first concern regards the terminology used in the manuscript. "Occupancy" is a well-defined term in structural biology, used in atomic models of biomolecules to describe the fraction of molecules that contain a particular group of atoms. It is typically used in two situations: to model alternative conformations (in which multiple copies of a given atom are included in the model, with occupancies that sum to 1) or to describe partial occupancy, in which a group of atoms have occupancy less than 1 to indicate that they are not present in all molecules being considered. In cryo-EM, this would mean the atoms are present in only a fraction of the particle images in the data set, and this is the meaning that relates most closely to the way the authors use "occupancy" in this manuscript. However, there is an important difference: in an atomic model (in the absence of damage) the occupancy should be the same for all atoms in a single connected molecule (e.g. a cofactor or a protein chain), since we know that if part of the molecule is present, the rest of it should be too, even if it is too poorly resolved to trace it in the density. By contrast, in this work, the "occupancy" can and does vary between different parts of the same protein chain. (This is seen in all of the results from real cryo-EM data, figs 2-5.)

In the authors' defence, I should note that the existing "LocOccupancy" method that they mention suffers from the same confusion. There is also still some contention in the wider structural biology community about how poorly-resolved atoms should be represented in models, and one of the "standard" ways is to include the atoms with low or zero occupancy values. This usage is much closer to the way the authors use "occupancy" in this manuscript. However, advocates of this approach are in a minority and it is certainly a contentious enough point that the authors should not just use the term in this way without comment. I suggest that the authors add a paragraph to the manuscript to explain exactly what they mean by the term "occupancy", particularly for structures that are flexible or have amorphous regions, and how it relates to the meaning of occupancy in atomic models. It might also be helpful if they could distinguish more clearly between situations where the occupancy is truly important and those where estimating and controlling for heterogeneity from any source (i.e. occupancy, flexibility or other) is sufficient.

My second concern is about the sources of signal attenuation in cryo-EM reconstructions. On page 6, the authors state that "local resolution variability within published reconstructions is common and

primarily due to flexibility". However, this neglects another important source of variability, which is misalignment of the 2D particle images with respect to the 3D reconstruction (and I note that the reference [42] that the authors cite as a source for the quoted sentence does not mention misalignment at all, and as far as I can tell does not discuss relative contributions of different sources of local resolution variability in any detail). Angular misalignment causes a decrease in resolution with distance away from the centre of the box, even for completely rigid particles, and it is very common for cryo-EM reconstructions to display this pattern to a greater or lesser extent. I'm sure the authors are well aware of this, which makes me all the more surprised that they don't mention it at all. Indeed, they do state that decreased resolution away from the box centre is expected, but they imply it is always due to structural flexibility, which is simply not the case. Often both factors are involved, because peripheral regions are typically more flexible as well as being more subject to the effects of angular misalignment, but it is incorrect to say the loss of resolution is primarily due to flexibility. In most places in the manuscript where the authors mention "flexibility", they should really say "flexibility or misalignment". It would be helpful to have a fuller explanation of this point somewhere in the paper too.

One other minor concern is that the authors neglect to mention the effect of charge in their discussion of atoms of unequal mass. In cryo-EM, atoms that carry a charge are well known to display anomalous densities (e.g. negatively charged atoms can disappear completely at some resolutions). This is particularly pertinent to their nucleosome example in Fig. 3a, since the negatively charged phosphate groups might not appear as strongly in the density map as would be expected from their mass. It would be interesting to see the same test on an example containing nucleic acids at much higher resolution, where the phosphate density would likely be stronger. If no such example is available, the authors should consider adding a brief discussion of the effects of charge into this section of the manuscript.

Another concern is that in a few places the authors make overly strong statements or over-state their conclusions. For example:

- In the abstract, "our implementation can derive reliable input parameters automatically" and on page 9, "we find OccuPy to be a reliable tool available to quantitatively estimate macromolecular occupancy within cryo-EM reconstructions" - saying "reliable" and "automatically" implies that it is fool-proof, but the results demonstrate that user expertise and thought is required in many cases (e.g. the detergent micelle example). These statements should be qualified, e.g. "when used with care, we find OccuPy to be a reliable tool..."

- In the abstract, "the resulting estimate is accurate" - I'm not sure this has been well enough justified. The authors could reasonably say the estimates are self-consistent, at least when the data are not too noisy, but "accurate" is too strong in the absence of some stronger evidence to justify it (which could be provided, for example, by a more realistic test on synthetic data including effects of noisy images, misalignment and flexibility)

In the introduction (top of p2), "it is ... prudent ... to investigate if and how unsupervised methods could improve [reconstruction] outcomes" - this is true, but potentially misleading. As mentioned above, OccuPy seems to require user input and expertise and so it is actually more of a supervised method, even if it could in principle be used in an unsupervised way. Perhaps the authors should just remove the word "unsupervised" here?

- End of p2, "illustrates that very small kernel sizes can be used without significant detriment to the estimate" - this has only been shown using synthetic example data with no flexibility or misalignment. The authors should add an appropriate caveat to this sentence to avoid readers getting the impression that small kernel sizes can be safely used with real data.

There are a few parts of the manuscript that are hard to understand and could be written more clearly:

- In the introduction, the sentence that begins "Since reference bias confines convergence"
- The final sentence of the discussion, beginning "OccuPy nonetheless represents"
- In the introduction, the explanation of "occupancy mode" is somewhat buried and hard to follow.

Given how much the occupancy mode features elsewhere in the manuscript, it would be very helpful to introduce it more clearly by explaining what its purpose is and briefly how it works.

Finally, there are a number of grammar and spelling mistakes. The manuscript would benefit from careful proof-reading and editing before publication. Here is a list of mistakes that I found (but there may also be others that I missed):

- p1: Methods for isolating homogeneous subsets of particle images through e.g. clustering is therefore employed, and commonly utilizes gradient descent methods -> are therefore employed, and commonly utilize gradient descent methods
- p2: The imperative to minimize over-fitting has lead to methods that avoid detrimental reference bias, and some which implicitly isolates useful reference bias -> has led to methods ... which implicitly isolate useful reference bias
- p2: Higher quality data and more robust reconstruction methods has also lead to more frequent iterative updates -> have also led to more frequent iterative updates
- p2: Reducing the kernel size does mitigates this effect -> does mitigate this effect
- p9: S is thus a normalized estimate of the of the local signal strength. We estimates S ... -> S is thus a normalized estimate of the local signal strength. We estimate S ...
- There are also some failed cross references, rendered as "Fig. ??"

Reviewer #2:

Remarks to the Author:

The authors introduce an interesting algorithm to compensate for different intensities in the 3D reconstruction of cryoEM maps. Different occupancies, local heterogeneity, or resolution may cause these differences. The authors study the properties of their algorithm under different scenarios, showing the usefulness of their approach.

The algorithm to estimate the local scale is straightforward but apparently effective. However, once the local scale is determined, its correction seems an art. Different modifications (gamma and sigmoidal corrections or clipping) are suggested. This leaves the door open to somewhat subjective modifications. It would be great if the authors could provide some canonical way to correct the local scale (although the suggested modifications could be still available to explore other possibilities).

The manuscript would benefit from the following changes:

- It would be useful to show slices from the modified maps in Figs. 2, 3, 4, and 5 so that they can be compared to the original slices.
- There is a reference to a figure missing in page 11 (after Eq. 13).
- The retention of solvent is proposed as a possibility, but no example is given.
- Can the program be called from a command line? It would be very useful for its integration into scripts and workflows.

Reviewer comments

NCOMMS-23-14047

Reviewer concerns have been numbered to make referencing easier.

Rebuttal or responses are indicated in blue.

Changes to the manuscript are highlighted in green.

Reviewer #1 (Remarks to the Author):

Cryo-EM reconstructions often display variable local resolution: molecular density is better resolved in some parts of the reconstruction and worse in others. This can be caused by heterogeneity of the sample (both in composition, aka occupancy, and conformation, aka flexibility) and inaccuracies in the reconstruction process (misalignment of particle images or incorrect estimates of other parameters). Identifying these effects is helpful for characterising a reconstruction, and adjusting for them can potentially help to improve the quality of the reconstruction.

In this manuscript, Forsberg, Shah and Burt aim to both estimate and adjust for the effect of heterogeneity in a cryo-EM reconstruction through the development of a new algorithm and software tool that they call OccuPy. The authors have done well to develop a new algorithm that appears to be effective and is much faster than any existing alternative approaches. This is an important problem and they have made some useful progress. As they say, this is a first step and the real potential of their approach would be if it could be used automatically as part of an iterative refinement and reconstruction process. However I expect this initial development will already be quite helpful to the community, particularly for scientists with challenging data sets and complex workflows that rely on detailed classification and signal subtraction approaches.

1.0

We sincerely thank the author for taking the effort to understand and emphasize the context of our work entirely, and highlighting both the current and future impact of our work whilst still recognizing its deficiencies and presenting those in a thoroughly constructive manner.

The authors generally do a good job of describing their work and the results broadly support the conclusions. However I do have concerns about some aspects of how the work is presented and so I recommend revision before the manuscript is accepted for publication.

1.1

My first concern regards the terminology used in the manuscript. “Occupancy” is a well-defined term in structural biology, used in atomic models of biomolecules to describe the fraction of molecules that contain a particular group of atoms. It is typically used in two situations: to model alternative conformations (in which multiple copies of a given atom are included in the model, with occupancies that sum to 1) or to describe partial occupancy, in which a group of atoms have occupancy less than 1 to indicate that they are not present in all molecules being considered. In cryo-EM, this would mean the atoms are present in only a fraction of the particle images in the data set, and this is the meaning that relates most closely to the way the authors use “occupancy” in this manuscript. However, there is an important difference: in an atomic model (in the absence of damage) the occupancy should be the same for all atoms in a single connected molecule (e.g. a cofactor or a protein chain), since we know that if part of the molecule is present, the rest of it should be too, even if it is too poorly resolved to trace it in the density. By contrast, in this work, the “occupancy” can and does vary between different parts of the same protein chain. (This is seen in all of the results from real cryo-EM data, figs 2–5.)

In the authors’ defence, I should note that the existing “LocOccupancy” method that they mention suffers from the same confusion. There is also still some contention in the wider structural biology community about how poorly-resolved atoms should be represented in models, and one of the “standard” ways is to include the atoms with low or zero occupancy values. This usage is much closer to the way the authors use “occupancy” in this manuscript. However, advocates of this approach are in a minority and it is certainly a contentious enough point that the authors should not just use the term in this way without comment. I suggest that the authors add a paragraph to the manuscript to explain exactly what they mean by the term “occupancy”, particularly for structures that are flexible or have amorphous regions, and how it relates to the meaning of occupancy in atomic models. It might also be helpful if they could distinguish more clearly between situations where the occupancy is truly important and those where estimating and controlling for heterogeneity from any source (i.e. occupancy, flexibility or other) is sufficient.

The reviewer raises an important point, of which we are aware and will gladly exert effort to clarify. As the reviewer suggests, we use occupancy in the statistical sense, i.e. to attribute heterogeneity to a binary (i.e. present in its entirety or not) component of the set of particles used for the reconstruction. This is due to two principal reasons:

1. It agrees with its use in crystallography, as the fraction of the imaged sample that contains the relevant atoms in the modeled configuration
2. It signifies a meaningful interpretation of composition heterogeneity, with a direct relation to current processing practices and the parlance of the cryo-EM field.

As the reviewer suggests, occupancy traditionally annotates atoms of a model. Insofar as the reconstruction is a representation of the molecular structure we argue that it is pertinent to assess “occupancy” without the presence of an atomic model. We thus determine occupancy as a scalar field that annotates the scalar field reconstruction, rather than a point-wise annotation

of the point-wise model. Thus, partially structured regions may display partial occupancy which varies across a protomer that one would otherwise expect to have the same partial occupancy based on connectivity. Unfortunately this agrees with the contentious practice of annotating atoms of disordered regions as having zero occupancy, which we do *not* encourage. Ideally, OccuPy would be able to locally decompose flexibility and occupancy, which would likely demarcate the difference between disorder and occupancy as field-annotation of heterogeneity. We have done our best to highlight these aspects by revising the main text accordingly:

Introduction:

OccuPy also provides a method to reduce the influence of the former to better approximate macromolecular occupancy generalized as a scalar field (see discussion).

Discussion:

This work defines local scale as an estimate of relative contrast in cryo-EM reconstructions, which is assumed to be proportional to heterogeneity in the data used. In the absence of flexibility we further interpret this as occupancy, signifying a mixing parameter of binary composition inherent to the input data. This is consistent with the accepted definition of occupancy in structural biology, where it annotates the relative occurrence of atoms in a model that best agrees with the map on which it is based, whereas our interpretation annotates the map itself. This permits quantification where atoms cannot be distinguished, but also leads to ambiguities in regions of partial disorder. It should thus be clarified that field-annotation of occupancy as a mixing parameter of binary origin is only relevant to the extent that the underlying heterogeneity is in fact binary. The local scale is a natural generalization of occupancy (to heterogeneity more broadly) where the origin is non-binary. It is nonetheless informative to attempt to decompose the local scale as originating in either binary or continuous heterogeneity. The occupancy-mode local scale implemented here attempts to omit the latter to render the local scale maximally interpretable as a mixing component of binary heterogeneity, but depending on the nature of the underlying heterogeneity this may not be possible without ambiguity. This should be considered in interpreting the local scale output by OccuPy.

1.2

My second concern is about the sources of signal attenuation in cryo-EM reconstructions. On page 6, the authors state that “local resolution variability within published reconstructions is common and primarily due to flexibility”. However, this neglects another important source of variability, which is misalignment of the 2D particle images with respect to the 3D reconstruction (and I note that the reference [42] that the authors cite as a source for the quoted sentence does not mention misalignment at all, and as far as I can tell does not discuss relative contributions of different sources of local resolution variability in any detail). Angular misalignment causes a decrease in resolution with distance away from the centre of the box, even for completely rigid particles, and it is very common for cryo-EM reconstructions to display this pattern to a greater or lesser extent. I’m sure the authors are well aware of this, which makes me all the more surprised that they don’t mention it at all. Indeed, they do state that decreased resolution away from the box centre is expected, but they imply it is always due to

structural flexibility, which is simply not the case. Often both factors are involved, because peripheral regions are typically more flexible as well as being more subject to the effects of angular misalignment, but it is incorrect to say the loss of resolution is primarily due to flexibility. In most places in the manuscript where the authors mention “flexibility”, they should really say “flexibility or misalignment”. It would be helpful to have a fuller explanation of this point somewhere in the paper too.

The reviewer is correct. We have amended the main text throughout to clarify that misalignment will introduce loss of resolution.

1. “...e.g. due to flexibility, misalignment, and/or partial occupancy.”
2. “...partial occupancy is qualitatively well estimated in the absence of flexibility or other sources of variable resolution.”
3. “...including heterogeneity, flexibility, misalignment, and amorphous regions...”
4. “...and primarily due to flexibility[42] and other sources of misalignment[24].”
5. “Local scale estimates in the presence of flexibility or misalignment”
6. “...principally depends on macromolecular flexibility and occupancy (as well as particle misalignment)...”
7. In practice, contrast is attenuated through local flexibility and/or partial occupancy, as well as the inclusion of misaligned or bad particles.”
8. “...large variations in local resolution due to flexibility and imperfect particle image parameters”.

1.3

One other minor concern is that the authors neglect to mention the effect of charge in their discussion of atoms of unequal mass. In cryo-EM, atoms that carry a charge are well known to display anomalous densities (e.g. negatively charged atoms can disappear completely at some resolutions). This is particularly pertinent to their nucleosome example in Fig. 3a, since the negatively charged phosphate groups might not appear as strongly in the density map as would be expected from their mass. It would be interesting to see the same test on an example containing nucleic acids at much higher resolution, where the phosphate density would likely be stronger. If no such example is available, the authors should consider adding a brief discussion of the effects of charge into this section of the manuscript.

We agree that charge is relevant to consider in this context, and have expanded the main text to reflect this. We also note that if charge manifests as increased mass, this can be considered without special consideration for charge. Given this, we would argue that the much stronger relative scattering of the sulfur atom in Fig 3b-d exemplifies the potential issue at higher resolution, and how the normalization procedure in OccuPy to account for such variations in scattering. Further to this, we note that both EMD-13015 (at 3.0Å) in Figure S5 and EMD-3943 (at 3.0Å) in Fig S6 are higher-resolution representations that contain ribonucleic acid, neither of which display an emphasis on phosphate atoms.

“OccuPy is not equipped to consider the expected average density, charge, mass, or other causes for altered scattering potential of constituent atoms, nor a physical image formation model.”

1.4

Another concern is that in a few places the authors make overly strong statements or over-state their conclusions. For example:

1.4.1

- In the abstract, “our implementation can derive reliable input parameters automatically” and on page 9, “we find OccuPy to be a reliable tool available to quantitatively estimate macromolecular occupancy within cryo-EM reconstructions” - saying “reliable” and “automatically” implies that it is fool-proof, but the results demonstrate that user expertise and thought is required in many cases (e.g. the detergent micelle example). These statements should be qualified, e.g. “when used with care, we find OccuPy to be a reliable tool...”

The main text has been amended to reflect the reliance on user input.

1. *“our implementation can derive ~~reliable~~ reasonable input parameters ~~automatically~~”*
2. *“we find that OccuPy is the only tool able to ~~to be a reliable tool available to~~ quantitatively estimate macromolecular occupancy within cryo-EM reconstructions and ~~the only tool able to modify this occupancy in~~ a meaningful way, but that user intervention may be necessary to assure fidelity in this process.”*

1.4.2

- In the abstract, “the resulting estimate is accurate” - I’m not sure this has been well enough justified. The authors could reasonably say the estimates are self-consistent, at least when the data are not too noisy, but “accurate” is too strong in the absence of some stronger evidence to justify it (which could be provided, for example, by a more realistic test on synthetic data including effects of noisy images, misalignment and flexibility)

We agree that the present work only establishes the self-consistency of OccuPy against real data (through Figure S3 and S4), where no ground truth exists. However, we would argue that the validation against synthetic data in Figure 1 does establish that the estimation of composition heterogeneity from contrast loss using the methods implemented in OccuPy is accurate. We have however amended the manuscript to instead state that

“We show that our implementation can derive ~~reliable~~ reasonable input parameters ~~automatically~~, that composition heterogeneity can be estimated based on contrast loss, and the reconstruction can be modified accordingly to emulate altered constituent occupancy, ...”

1.4.3

-In the introduction (top of p2), “it is ... prudent ... to investigate if and how unsupervised methods could improve [reconstruction] outcomes” - this is true, but potentially misleading. As mentioned above, OccuPy seems to require user input and expertise and so it is actually more of a supervised method, even if it could in principle be used in an unsupervised way. Perhaps the authors should just remove the word “unsupervised” here?

Duly amended, the reviewer is correct.

“...more prudent than ever to investigate if and how ~~unsupervised methods could improve its outcomes could be improved~~ through supplementing relevant reference bias...”

1.4.4

- End of p2, “illustrates that very small kernel sizes can be used without significant detriment to the estimate” - this has only been shown using synthetic example data with no flexibility or misalignment. The authors should add an appropriate caveat to this sentence to avoid readers getting the impression that small kernel sizes can be safely used with real data.

We agree that it is pertinent to anticipate how users might misconstrue our findings, and have duly qualified this statement.

“...and illustrates that the ~~very small~~ kernel size ~~can be used~~ need only encompass a few voxels without significant detriment to the estimate when the fidelity and sampling of the underlying data is sufficiently high.”

1.5

There are a few parts of the manuscript that are hard to understand and could be written more clearly:

1.5.1

- In the introduction, the sentence that begins “Since reference bias confines convergence”

Amended to

“Since convergence of e.g. data clustering is directed by reference bias for the cluster reference, it is justified to consider that modifications of the latter based on estimated heterogeneity might be useful to improve convergence[34, 38].”

1.5.2

- The final sentence of the discussion, beginning “OccuPy nonetheless represents”

Amended to

“OccuPy thus offers an example of how local spatial analysis can improve interpretation of cryo-EM reconstructions, which stands to be developed further to benefit future reconstruction analysis and refinement algorithms broadly.”

1.5.3

- In the introduction, the explanation of “occupancy mode” is somewhat buried and hard to follow. Given how much the occupancy mode features elsewhere in the manuscript, it would be very helpful to introduce it more clearly by explaining what its purpose is and briefly how it works.

Occupancy mode has been clarified in the introduction.

OccuPy also provides a method to reduce the influence of the former to better approximate macromolecular occupancy generalized as a scalar field (see discussion). This so-called occupancy mode is established by application of a low-pass filter to approximately neutralize the influence blur variation on differential local contrast. This isolates composition heterogeneity of the reconstruction, which can justifiably be modified ~~can then further be used to modify the input~~ to emulate reconstructions expected from more homogeneous image data.

1.6

Finally, there are a number of grammar and spelling mistakes. The manuscript would benefit from careful proof-reading and editing before publication. Here is a list of mistakes that I found (but there may also be others that I missed):

1.6.1

- p1: Methods for isolating homogeneous subsets of particle images through e.g. clustering is therefore employed, and commonly utilizes gradient descent methods -> are therefore employed, and commonly utilize gradient descent methods

Duly changed

1.6.2

- p2: The imperative to minimize over-fitting has lead to methods that avoid detrimental reference bias, and some which implicitly isolates useful reference bias -> has led to methods ... which implicitly isolate useful reference bias

Duly changed

1.6.3

- p2: Higher quality data and more robust reconstruction methods **has** also **lead** to more frequent iterative updates -> have also led to more frequent iterative updates

Duly changed

1.6.4

- p2: Reducing the kernel size does mitigates **s** this effect -> does mitigate this effect

Duly changed

1.6.5

- p9: S is thus a normalized estimate **of the** of the local signal strength. We estimates **S** ... -> S is thus a normalized estimate of the local signal strength. We estimate S ...

Duly changed

1.6.6

- There are also some failed cross references, rendered as "Fig. ??"

Duly changed

Reviewer #2 (Remarks to the Author):

The authors introduce an interesting algorithm to compensate for different intensities in the 3D reconstruction of cryoEM maps. Different occupancies, local heterogeneity, or resolution may cause these differences. The authors study the properties of their algorithm under different scenarios, showing the usefulness of their approach.

2.0

The algorithm to estimate the local scale is straightforward but apparently effective. However, once the local scale is determined, its correction seems an art. Different modifications (gamma and sigmoidal corrections or clipping) are suggested. This leaves the door open to somewhat subjective modifications. It would be great if the authors could provide some canonical way to correct the local scale (although the suggested modifications could be still available to explore other possibilities).

We have clarified in the main text that amplification and attenuation are canonical modifications that will at most equalize or remove partial occupancies, and which can thus be considered unsupervised and safe at any applied power. Power less than +infinity applied will thus only effect partial modification, which is a means for more conservative output. We have also clarified that sigmoid modification is more tenuous and dependent on user interaction, amenable to more tailored visualization and/or interactive processing.

“The present work has devised methods to modify a reconstruction proportionally to the estimated local scale by amplification or attenuation of partial occupancy, as described in the methods section and exemplified in Fig. 1d. These modifications will at most remove or equalize partial occupancy, are intended to be manually devised through the provided GUI but can be adjusted to decrease the extent of modification. ~~and will decrease macromolecular heterogeneity by either increasing or decreasing regions with intermediate occupancy mode local scale.~~ Consequently, these modifications at full or finite power are natural methods to modify the reference bias in numerous current cryo-EM processing tools completely or partially, respectively, ~~to establish conditions of convergence.~~ The GUI also permits a sigmoid modification that combines amplification and attenuation, tailored to user-interactive visualization and modification such as subtraction. Here we evaluate how amplification, attenuation and differential sigmoid modification of local scale manifest in reconstructions with features encountered during cryo-EM processing, including heterogeneity, flexibility, and amorphous regions such as inherently disordered detergent.”

The manuscript would benefit from the following changes:

2.1

- It would be useful to show slices from the modified maps in Figs. 2, 3, 4, and 5 so that they can be compared to the original slices.

A supplementary figure has been added displaying the slices of all modified panels shown in Figure 2. Figure 3 and 5 does not display any modified maps.

Added to caption Fig 2:

“Central slices of all modified densities displayed are shown in Fig. S5.”

2.2

- There is a reference to a figure missing in page 11 (after Eq. 13).

Duly changed:

“illustrated in Fig. S9a and exemplified in Fig. 2.”

2.3

- The retention of solvent is proposed as a possibility, but no example is given.

Solvent noise is always retained unless the user specifies differently. The section referred to by the reviewer is simply concisely stating the method by which this is retained, through use of the solvent-model confidence estimate. An example of this confidence estimate is shown in Fig. S7, although this does not display opt-in solvent flattening directly.

2.4

- Can the program be called from a command line? It would be very useful for its integration into scripts and workflows.

Yes, the implemented GUI is actually a means of generating the equivalent command that the user could then run on the command-line. The GUI can also report this command, should the user prefer to run it directly on the command-line. This is now explicitly mentioned in the text.

To permit easy integration with current processing pipelines, a command-line interface and python module is also provided.

Reviewers' Comments:

Reviewer #1:

Remarks to the Author:

The authors have suitably addressed all of my concerns, and I appreciate the very clear way they presented their changes in the response letter. I am now very happy for this work to be published, but I do have a couple of very minor further suggestions:

- "neutralize the influence blur variation" -> "neutralize the influence of blur variation"
- "It should thus be clarified that filed-annotation of occupancy as a mixing parameter of binary origin is only relevant to the extent that the underlying heterogeneity is in fact binary" - I'm afraid I don't understand what "filed-annotation" means here. Is it a typo for "fixed annotation", or something else? Perhaps the authors could find a more clear wording.

Reviewer #2:

Remarks to the Author:

The authors have significantly improved the manuscript and all my comments have been properly addressed.